# Agnathia-Otocephaly Complex Due to a De Novo Deletion in the *OTX2* Gene

**DOI:** 10.3390/genes13122269

**Published:** 2022-12-02

**Authors:** Marco Fabiani, Francesco Libotte, Katia Margiotti, Dina Khader Issa Tannous, Davide Sparacino, Maria Pia D’Aleo, Francesca Monaco, Claudio Dello Russo, Alvaro Mesoraca, Claudio Giorlandino

**Affiliations:** 1ALTAMEDICA, Human Genetics Laboratory, Viale Liegi 45, 00198 Rome, Italy; 2School of Medicine and Surgery, Department of Obstetrics and Gynecology, UniCamillus-Saint Camillus International University of Health Sciences, Via di Sant’Alessandro, 8, 00131 Rome, Italy; 3ALTAMEDICA, Department of Prenatal Diagnosis, Fetal-Maternal Medical Centre, Altamedica Viale Liegi 45, 00198 Rome, Italy

**Keywords:** OTX2, agnathia, otocephaly, clinical exome sequencing, de novo variant, synotia and proboscis, AOC

## Abstract

Agnathia-otocephaly complex (AOC) is a rare and usually lethal malformation typically characterized by hypoplasia or the absence of the mandible, ventromedial and caudal displacement of the ears with or without the fusion of the ears, a small oral aperture with or without a tongue hypoplasia. Its incidence is reported as 1 in 70,000 births and its etiology has been attributed to both genetic and teratogenic causes. AOC is characterized by a wide severity clinical spectrum even when occurring within the same family, ranging from a mild mandibular defect to an extreme facial aberration incompatible with life. Most AOC cases are due to a de novo sporadic mutation. Given the genetic heterogeneity, many genes have been reported to be implicated in this disease but to date, the link to only two genes has been confirmed in the development of this complex: the orthodenticle homeobox 2 (*OTX2*) gene and the paired related homeobox 1 (*PRRX1*) gene. In this article, we report a case of a fetus with severe AOC, diagnosed in routine ultrasound scan in the first trimester of pregnancy. The genetic analysis showed a novel 10 bp deletion mutation c.766_775delTTGGGTTTTA in the *OTX2* gene, which has never been reported before, together with a missense variant c.778T>C in cis conformation.

## 1. Introduction

Agnathia otocephaly complex (AOC) (MIM#202650) is a rare malformation typically characterized by hypoplasia or the absence of the mandible (agnathia), ventromedial and caudal displacement of the ears (melotia), with or without the fusion of the ears (synotia), a small oral aperture (microstomia), with or without a tongue hypoplasia or absence (aglossia) [1]. AOC has been associated with extra-facial features of which the most common is holoprosencephaly, a failure in the complete division of the two brain hemispheres [2]. A defect in blastogenesis or aberrancies in the neural crest cell migration can result in an incomplete development of the first pharyngeal arch, giving rise to an otocephalic association of abnormalities involving the ears, mouth, and mandible [3]. In most cases, the complex is lethal due to the difficulty in respiration and ventilation. The etiology has been attributed to both genetic and teratogenic causes [4]. Exposure during pregnancy, to salicylates, theophylline, radiation and alcohol have been reported amongst many others [1,5,6]. AOC was first described by Kerckring in 1717, and its incidence is reported as 1 in 70,000 births [5]. Despite most AOC cases are attributed to a de novo sporadic mutation, a few familial cases have, however been reported [7,8,9,10,11,12]. AOC is characterized by a wide severity clinical spectrum, even when occurring within the same family, ranging from a mild mandibular defect to an extreme facial aberration, incompatible with life [5,8,13,14]. Given the genetic heterogeneity, many genes have been reported to be implicated in this disease [8,15], for example the loss of the function of the *PGAP1* gene has been shown to cause otocephaly in a mouse model [16], but to date in humans, only two genes have been confirmed to have a direct link in the development of this complex disorder, paired related homeobox 1 (*PRRX1*; MIM*167420) and orthodenticle homeobox 2 (*OTX2*; MIM*600037) genes. As of today, although these genes have been associated with AOC, no genotype-phenotype correlation has been established [17]. We hereby report a case of a fetus with severe facial defects diagnosed on routine ultrasound scan in the first trimester of pregnancy, later confirmed to be a severe case of AOC. A genetic analysis showed that the fetus was heterozygous for a novel 10 bp deletion variant c.766_775delTTGGGTTTTA, causing a frameshift mutation in the *OTX2* gene, never before reported. In addition, together with the deletion variant, it was also identified as a missense variant c.778T>C in the same gene found in cis conformation.

## 2. Materials and Methods

Genomic DNA was extracted from the proband’s tissue sample after the pregnancy was terminated and from the parental peripheral blood using the DNeasy Blood & TissueKit and QIAamp DNA Blood Mini Kit (Qiagen, Hilden, Germany), according to the manufacturer’s instructions in the following link (https://www.qiagen.com/us/products/discovery-and-translational-research/dna-rna-purification/dna-purification/genomic-dna/dneasy-blood-and-tissue-kit/, accessed on 15 November 2022). The chromosome karyotyping analysis of the fetus was performed on cells harvested from the umbilical region with conventional G-banding, according to standard procedures. Sixteen metaphases were analyzed. The chromosome analysis of the parental blood samples was performed using GTG-banding techniques on the PHA-stimulated blood lymphocytes. The array comparative genomic hybridization (aCGH) analysis was performed using 44K platform (Agilent Technologies, Santa Clara, CA, USA) on DNA from the cultured fetus cells and DNA from the parental blood, to characterize the presence of the DNA deletions or duplications, as previously reported [18]. Clinical exome sequencing (CES) was carried out using the TruSight One Sequencing Panel (Illumina, San Diego, CA, USA), according to the manufacturer’s instructions. The panel covers 4813 disease-associated genes. The targeted exonic regions underwent paired-end sequencing on an Illumina platform, using a NextSeq 500 sequencing system (NextSeq High Output Kit, 300 cycles Illumina, San Diego, CA, USA). The data analysis variants were carried out with the Illumina Variant Studio software v3.0 and the BaseSpaceVariant Interpreter Beta (Illumina). The detected variants were annotated and filtered, based on the information of the functional prediction [e.g., Polyphen2 (http://genetics.bwh.harvard.edu/pph2/, accessed on 15 November 2022) SIFT (https://sift.bii.a-star.edu.sg/, accessed on 15 November 2022), REVEL (https://genome.ucsc.edu/cgi-bin/hgTrackUi?db=hg19&g=revel, accessed on 15 November 2022)], disease association [(e.g., ClinVar (https://www.ncbi.nlm.nih.gov/clinvar/, accessed on 15 November 2022), HGMD (https://www.hgmd.cf.ac.uk/ac/index.php, accessed on 15 November 2022), OMIM (https://www.omim.org/, accessed on 15 November 2022) and GWAS (https://www.ebi.ac.uk/gwas/, accessed on 15 November 2022)] and the population allele frequency [e.g., dbSNPs (https://www.ncbi.nlm.nih.gov/snp/, accessed on 15 November 2022), ALFRED (https://www.re3data.org/repository/r3d100012700, accessed on 15 November 2022)].

The identified variants in the *OTX2* gene were verified using Sanger sequencing with a set of primers (OTX2_F:5′-GGAATTTCCACGAGGATGTC-3′,OTX2_R:5′-CTACTTTGGGGCATGGACT-3′). The PCR was performed in a 50-μL reaction containing a final concentration of 1× PCR Buffer (Applied Biosystems, Foster City, CA, USA), 50 μmol/L each of dNTP, MgCl_2_ 1.5 mM, 1.25 AmpliTaq Gold (Applied Biosystems), and 0.2 μmol/L each forward and reverse primers. The reaction mixture was subjected to 95 °C for 5 min, followed by 35 cycles of 95 °C for 15 s, 57 °C for 15 s, and 72 °C for 1 min, followed by 72 °C for 7 min. The cycle sequencing was performed using the BigDye version 3.1 terminator cycle-sequencing kit, according to manufacturer’s instructions (Applied Biosystems). The cycle-sequencing conditions were 95 °C for 30 s, followed by 35 cycles of 95 °C × 15 s, 50 °C × 15 s, and 60 °C × 4 min. The products were analyzed using a SeqStudio Genetic Analyzer (Applied Biosystems). The PCR products were also identified on 3% agarose gel electrophoresis and 0.5× TAE gel (40 mM Tris–HCl, 20 mM acetic acid, 1 mM EDTA, pH 8.0), containing GelRed^®^ (1 μg/mL) for DNA staining and visualization. For the calculation of the DNA fragment, size a molecular-weight DNA size marker (1 Kb Plus DNA Ladder—Invitrogen^®^, Carlsbad, CA, USA) was included in each gel run. The selected DNA bands were cut out of the agarose gels with surgical blades and the excised gel was purified from the agarose gels and sequenced, according to the above described procedure.

The predicted structure and the protein features of the frameshift variant Leu256ThrfsTer43 on the *OTX2* protein was obtained using the trRosetta tool (https://yanglab.nankai.edu.cn/trRosetta/, accessed on 15 November 2022), whereas the predicted protein features (e.s DISULFIND) were calculated using the PredictProtein tool (https://predictprotein.org/, accessed on 15 November 2022).

## 3. Case Presentation

### 3.1. Clinical Features at Ultrasound and the Post Abortion Examinations

A 31-year-old pregnant woman primigravida, with no remarkable family history or consanguinity, underwent an ultrasound examination, as part of the regular first trimester screening at 13 weeks of amenorrhea. The ultrasound examination of the pelvis showed an increased uterine body volume, a normally implanted gestational sac, and a single live embryo with a crown rump length (CRL) of 57 mm; corresponding to the gestational age. Upon the 2D ultrasound scanning, using GE healthcare Voluson E10, the absence of the mandible (agnathia) and the asymmetry of the maxillary bones were noted. In addition, the eye orbits appeared misaligned, with evidence of hypotelorism, and a tubular bulge (proboscis) was evidently protruding from the lower part of the face (Figure 1). 

The pregnant woman and her husband (40 years-old) denied any known family history of inheritable genetic cond itions or exposure to teratogens, such as alcohol, medication or irradiation during pregnancy. Due to the severity of the malformations observed in the ultrasound, the couple decided to undergo a medical abortion. The post-mortem examination of the fetus revealed the severe malformation of the face, including agenesis of the mandible (agnathia), a small mouth (microstomia), ventromedial malpositioning of the ears (melotia) with the auricular fusion at the level of the neck (synotia), a single eye socket in the center of the face (cyclopia), absence of the tongue (aglossia), absence of the pharyngeal floor and the presence of proboscis (Figure 2). Extra facial features were present in the form of holoprosencephaly and talus valgus, however no malformations of the abdominal or thoracic organs were seen and situs inversus, a recurring feature in AOC cases, was not detected. 

### 3.2. Genetic Analysis

Following the medical abortion, fetal karyotyping was performed; the quantitative fluorescent polymerase chain reaction (QF-PCR) and a cytogenetic examination showed a normal male karyotype 46,XY. In order to exclude the chromosomal rearrangement due to the loss or gain of a small quantity of genetic material, an array comparative genomic hybridization (aCGH) technique was also performed. The results of the aCGH were negative, indicating no gain or loss of genetic material in the fetus. Subsequent to the aCGH, a clinical exome sequencing (CES) by Next Generation Sequencing (NGS) (Illumina Next-Seq500) was performed. By using the prioritization bioinformatic tool Exomiser (https://www.sanger.ac.uk/tool/exomiser/, accessed on 1 August 2022) on the CES results, we were able to prioritize for our phenotype 2 heterozygous variants in the *OTX2* gene: NM_021728.3:c.766_775delTTGGGTTTTA and NM_021728.3:c.778T>C (Figure 3A). Using Sanger sequencing, the variants were further verified as heterozygous variants in the proband (Figure 3B). Moreover, Sanger sequencing extended to the parents showed the absence of both variants, thereby indicating that the mutations occurred de novo (Figure 3B). Moreover, we investigated the variants located in the genes or the modifying factors likely implicated in the AOC phenotype, such us *PRRX2* (MIM*604675), *TWIST1* (MIM*601622), *NDST1* (MIM*600853), *BMP4* (MIM*11 2262), FGF8 (MIM*600483), *ALX4* (MIM*605420), *MSX1* (MIM*142983), *PGAP1* (MIM*611655), *CNBP* (MIM*116955) and *TWSG1* (MIM*605049), ref. [9,19], but no potentially pathogenic or VUS variants were found. 

When running the wild type (in this case the mother of the affected fetus) PCR product on 3.5% agarose gel, we detected a single amplified DNA fragment of about 290 bp, as expected from a normal allele, whereas the PCR product from the fetus DNA showed two bands comprising an aberrant band (≈280 bp), that was smaller than that of the mother, likely corresponding with the identified 10 bp deletion in the OTX2 gene (Figure 4).

Genetic online databases were reviewed for the presence of the variants identified in this case. The c.778T>C variant was not present on the ClinVar Database (https://www.ncbi.nlm.nih.gov/clinvar/ accessed on 1 September 2022), but it was present on the Varsome Database (https://varsome.com/, accessed on 15 November 2022). The ACMG Classification reported PP3 and PM2 criteria and the following ACMG standards and guidelines for the interpretation of the variants [doi:10.1038/gim.2015.30] it can classify the variant as a variant of uncertain significance (VUS) (see ACMG Classification section at https://varsome.com/variant/hg38/NM_021728%3Ac.778T%3EC?, accessed on 15 November 2022) [20]. No data are present about the frequencies of this variant in any population. Furthermore, we reviewed the mutation pathogenicity prediction software [Mutation Taster (https://www.mutationtaster.org/, accessed on 15 November 2022), Polyphen2 (http://genetics.bwh.harvard.edu/pph2/, accessed on 15 November 2022), Predict SNP (https://loschmidt.chemi.muni.cz/predictsnp/, accessed on 15 November 2022), FATHMM (http://fathmm.biocompute.org.uk/, accessed on 15 November 2022), PhD-SNP (https://snps.biofold.org/phd-snp/phd-snp.html, accessed on 15 November 2022) and CADD (https://cadd.gs.washington.edu/, accessed on 15 November 2022)], which predicted the mutation c.778T>C as pathogenic or deleterious. Despite c.778T>C is a missense mutation of Ser260Pro with a probably damaging or disruptive function on the protein structure, in cis conformation with the c.766_775delTTGGGTTTTA variant, causes a downstream mutation c.769T>C on the frameshifted allele, leading a synonymous variant Thr257Thr, without any clinical significance. Nonetheless, the c.766_775delTTGGGTTTTA was a novel variant not reported to date in any of the examined population databases [Clinvar (https://www.ncbi.nlm.nih.gov/clinvar/, accessed on 15 November 2022), Varsome (https://varsome.com/, accessed on 15 November 2022), 1000 Genome Project (https://www.internationalgenome.org/, accessed on 15 November 2022), HGMD (https://www.hgmd.cf.ac.uk/ac/index.php, accessed on 15 November 2022), ExAC (https://exac.broadinstitute.org/, accessed on 15 November 2022), and gnomAD (https://gnomad.broadinstitute.org/, accessed on 15 November 2022)]. The variant has no frequency, neither in gnomAD nor in other population databases. As for the variant c.766_775delTTGGGTTTTA (Leu256ThrfsTer43), it is a deletion of 10 bases in exon 5 causing a frameshift deletion, resulting in a different C terminal protein from aa 256 till the end after 43 aa, generating an abnormal protein product. Despite the frameshift deletion of our case, it does not impact the DNA binding domain or the regulation domains of the *OTX2* protein, we notice that the predicted structure of the *OTX2* protein bearing the c.766_775delTTGGGTTTTA deletion, caused a loss of the Disulfide bond predicted by the DISULFIND server (https://bio.tools/disulfind, accessed on 15 November 2022) and used the PredictProtein tool (https://predictprotein.org/, accessed on 15 November 2022) (Figure 5A), to induce the stretching of the protein structure (Figure 5B) [21]. 

## 4. Discussion

AOC is a lethal malformation that has been the first course and caliber of the carotid artery, and has been observed in patients with AOC, therefore it was diagnosed in an ultrasound scan in 1977 [22]. AOC is diagnosed by the presence of an underdeveloped or absent mandible, the malpositioning of the ears with or without the auricular fusion and a small mouth with oroglossal hypoplasia or aglossia. Other features may also be present and can greatly alter the prognosis, these include holoprosencephaly, cyclopia, anencephaly, meningomyelocele, situs inversus, lung and genitourinary anomalies, and skeletal and cardiovascular anomalies [1]. Abnormalities in the face, some literature suggests that the hypoperfusion to the face and head may play a role in the development of brain malformations in these patients, as it may possibly cause the secondary loss of the craniofacial structures [5]. The prognosis of AOC fetuses is extremely poor, and long-term survival is almost impossible. Only a small number of individuals live after birth, with less than 5% surviving beyond two months of life [1]. Almost all infants die of respiratory distress due to the absence of the mandible hindering the development of the adjacent structures. This often involves the naso-mandibular complex and the oropharynx, causing, other than respiratory distress, feeding difficulties, and speech and/or hearing impairment. Those who survive still require a tracheostomy for assisted breathing and a gastrostomy tube for assisted feeding [23]. Longer-term survival depends on the severity of the associated malformations, as of today only one case of mild dysgnathia survived till adulthood with mild to moderate hearing loss [8]. 

The diagnosis of AOC from the ultrasound scan proves to be a challenge particularly in the first trimester. In the absence of other associated extra-facial features, diagnosis may be delayed or missed. This itself poses a psychological burden on the parents who are faced with the difficult decision of abortion versus continuing with a pregnancy that is likely to have a non-viable outcome. The majority of cases are diagnosed in the second trimester, however, in cases diagnosed at specialized fetal medicine units, diagnosis is possible in the first trimester [23]. AOC does not show any predilection to females or males and, as mentioned previously its etiology could be both of genetic or teratogenic origin. In animal models, agnathia otocephaly has been seen spontaneously in teratogen exposure e.g., irradiation, hyperthermia, trypan blue and streptonigrin [1,24,25]. Furthermore, case reports in humans have demonstrated cases of AOC associated to the exposure to salicylates, alcohol, radiation, oxymetazoline, mycophenolate, phenytoin, amidopyrine and beclomethasone [1,5,6,26]. 

Genetic diagnosis of AOC is arduous, due to the rarity of this disease and the genetic heterogeneity. Four cases of an unbalanced translocation of der (18),t (6,18) (pter->p24.1 or p24.2::p11.21->qter) have been reported as an inherited aberration from a balanced translocation in a parent [6,11]. However, the genetic causes are principally assigned to a point mutation in the *PRRX1* and *OTX2* genes [2,9,14,15]. Animal models have suggested the involvement of other genes in AOC, including *PGAP1*, *ALX4*, *MSX1* and *TWSG1* genes. A mouse has investigated the *CNBP* gene as a modifier gene, suggesting a potential role in AOC, despite a following human study in a cohort of 10 AOC cases that did not reveal any mutation in the *CNBP* gene suggesting that mutations in *CNBP* might not be involved in such a phenotype in humans [2]. As of today, the human genetic association to AOC has been confirmed only for the *PRRX1*, and *OTX2* genes [9]. The *PRRX1* gene encodes a nuclear homeobox protein that functions as a transcription co-activator, enhancing the DNA-binding activity of the serum response factor, a protein required for the induction of genes by growth and differentiation factors. The *PRRX1* protein regulates muscle creatinine kinase, indicating a role in the establishment of diverse mesodermal muscles during embryogenesis and regulates the formation of the pre-skeletal condensations from undifferentiated mesenchyme [8]. The *PRRX1* gene mutation was first described in 2011, by Sergi et al. [19] and it has shown the autosomal dominant and recessive inheritance. Such mutations account for less than 15% of AOC cases reported [2]. The *OTX2* gene encodes a transcription factor with essential functions in the embryonic head formation and, at later developmental stages, in the eye and brain development. Following birth, although the *OTX2* expression decreases, it maintains an important role in the retina [15]. Mouse models confirmed the involvement of *OTX2* in the pharyngeal arch formation and how its disruption leads to major craniofacial malformations [9]. Heterozygous *OTX2* mutants were found to display AOC phenotypes in variable proportions, depending on the genetic background of the mice, suggesting a role of a modifier gene. These mutants also inconstantly displayed anophthalmia and holoprosencephaly [27]. The *OTX2* mutations have been associated with agnathia otocephaly, but also other severe phenotypes, such as syndromic type 5 microphthalmia, pituitary hormone deficiency and the early onset of retinal dystrophy [2,9]. Recent large data from human genome sequencing studies presented in the gnomAD database, showed that the pLI score (the probability of loss-of-function intolerance) for *OTX2* is >0.9, meaning that *OTX2* falls into the class of the loss-of-function haploinsufficient genes. Most of the *OTX2* variants identified in patients, lead to a truncated or absent protein and thus to the *OTX2* haploinsufficiency [14]. Indeed Patat and colleagues presented a case of AOC with agnathia, astomia and aglossia, an absent pharyngeal floor, low posteriorly rotated, paramedian and convergent ears with a nonsense variant of p.Arg97* at the *OTX2* transcript [14]. Due to the strong effect of the mutation, it was hypothesized that a loss-of-function variant affecting the coding sequence of a gene, most likely would be responsible for the disease. 

In this article, we report a case of agnathia otocephaly with holoprosecephaly, with a novel de novo variant in the *OTX2* gene, c.766_775delTTGGGTTTTA (Figure 3A), together with a further c.778T>C missense variant in cis conformation. The c.766_775delTTGGGTTTTA causes a frameshift mutation in the *OTX2* gene and refers to deletion of the nucleotide bases in numbers that are not multiples of three, at the end of the *OTX2* protein, causing a stop codon after 43 aminoacids and an abnormal protein product. Regarding the other identified variant, the c.778T>C results located three bases upstream of the c.766_775delTTGGGTTTTA deletion, are unlikely to attribute a pathogenicity to this variant, since the deletion will further change and the predicted amino acid is a synonym substitution. 

Several public genetic databases (Clinvar, HGMD, Varsome, GnomAD) reported frameshift mutations in the *OTX2* gene, such as c.781_784del (p.Thr261fs), c.811del (p.Thr271fs), c.634_640del (p.Tyr212fs) c.673del (p.Ala225fs), and c.698del (p.Asn233fs), causative of an eye defect, as Leber congenital amaurosis or syndromic microphthalmia type 5, without any other craniofacial defect [28].

The transcriptional reporter assays demonstrated that the OTX2 proteins lacking the homeodomain are inactive, whereas those lacking the C-terminal domain, as in the case reported here, result in an 80% reduction in the transactivation [29]. Despite the frameshift deletion of c.766_775delTTGGGTTTTA, that does not impact the DNA binding domain or the regulation domains [21] that predict the structure of the *OTX2* protein bearing the c.766_775delTTGGGTTTTA deletion, it is a cause of a loss of the Disulfide bond (predicted by DISULFIND) (Figure 5B), causing a probable stretching of the protein structure (Figure 5A). This anomaly of the tertiary protein structure might explain the AOC phenotype. 

## 5. Conclusions

AOC remains a rare malformation and our understanding of its etiology is still growing. The variability of the expression and inheritance of AOC suggests a multifactorial, multi-gene involvement that is not yet fully understood. The degree of involvement of each genetic mutation is difficult to evidence, with such a small number of reported cases. In this case, we identified two de novo variants in cis and we hypothesize that the variant c.766_775delTTGGGTTTTA is the likely culprit for this extreme agnathia otocephaly phenotype. The mutation of c.766_775delTTGGGTTTTA causes a complete disruption of the *OTX2* protein, probably modifying not only the structure the C-terminal, but also the whole tertiary structure of the *OTX2* protein. We could also exclude a combination effect given by c.778T>C, since it causes the in cis conformation with c.766_775delTTGGGTTTTA, which is a synonymous variant. With the spread of genetic technology and the improved access to genetic testing, we hope to see the increased reporting of genetic mutations in AOC, and develop a better understanding of their significance.

Awareness of AOC is important, particularly amongst gynecologists and scanning physicians. Performing routine 2D scanning of the fetal profile during first trimester scan, is an effective screening tool and the use of 3D imaging should also be considered in all suspected cases, when available, to confirm the diagnosis [30]. An early diagnosis allows for better counselling of the parents and provides them with the appropriate time to digest the information and appreciate the severity of the situation, before embarking on a cumbersome decision, such as medical abortion. 

## Figures and Tables

**Figure 1 genes-13-02269-f001:**
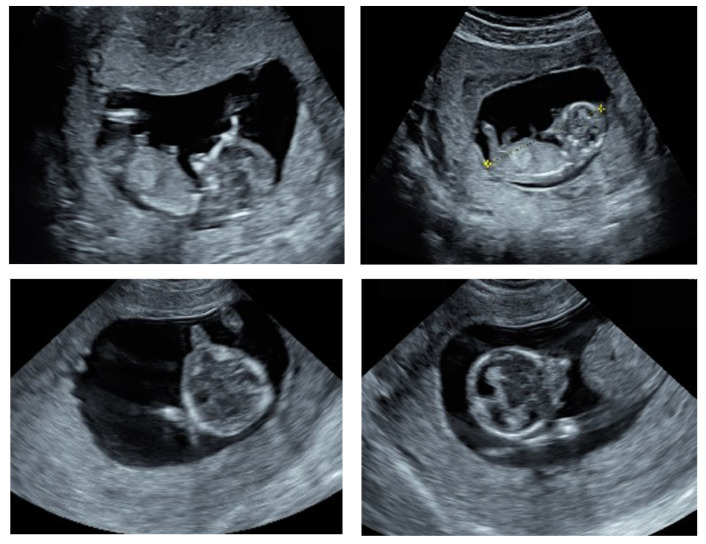
Ultrasound findings of the fetus at 11 weeks gestation. Sagittal sections (**top two images**) showing the absence of the mandible and the axial sections (**lower two images**) showing the asymmetry of the maxillary bones and a protruding tubular structure (proboscis) from the lower part of the face.

**Figure 2 genes-13-02269-f002:**
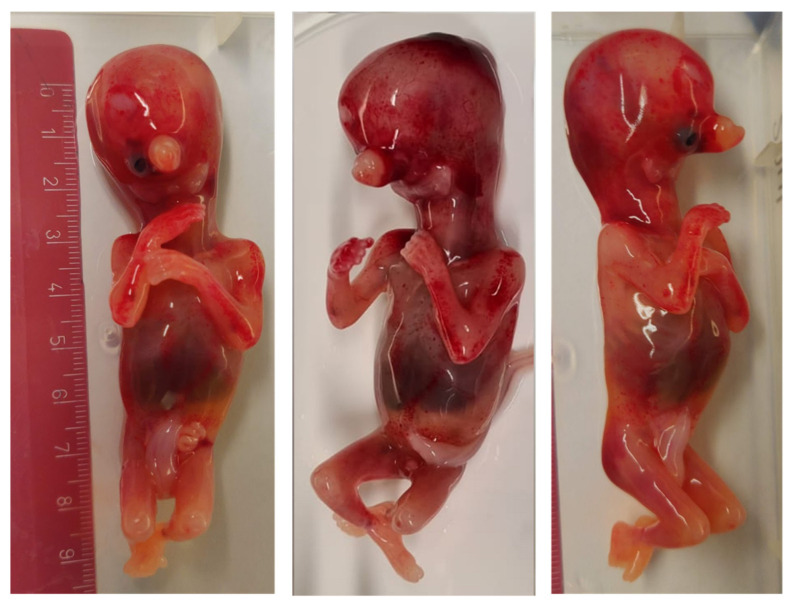
Front and profile of the fetus at 13 weeks after a medical abortion, showing cyclopia, agnathia, synotia and the presence of a proboscis.

**Figure 3 genes-13-02269-f003:**
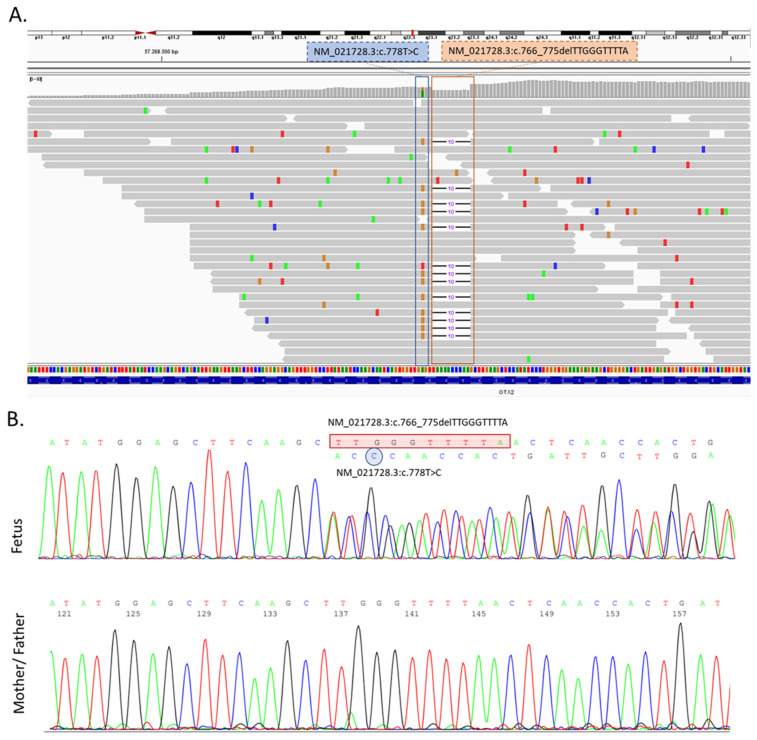
(**A**) Visualization of the result of the clinical exome sequencing (CES) showing the identified variant in the *OTX2* gene by Integrative Genomics Viewer (IGV) software (https://software.broadinstitute.org/software/igv/ accessed on 1 August 2022). c.778T>C and c.766_775delTTGGGTTTTA variants were in a heterozygous status in the proband. (**B**) Sanger sequencing confirmed the identified *OTX2* variants, which were not present in his parents.

**Figure 4 genes-13-02269-f004:**
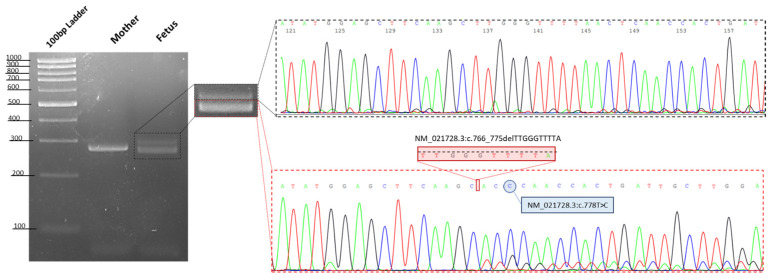
Gel image showing the PCR products of exon 5 of the *OTX2* gene. The resultant PCR products using DNA from a control (in this case the mother of fetus), showed a single amplified DNA fragment of 290 bp, as expected of a normal allele, whereas the proband showed two bands comprising an aberrant band (280 bp), that was smaller than that of the mother. Sanger sequencing shows the two different alleles in the proband, highlighting the presence in cis of the two identified variants in the *OTX2* gene.

**Figure 5 genes-13-02269-f005:**
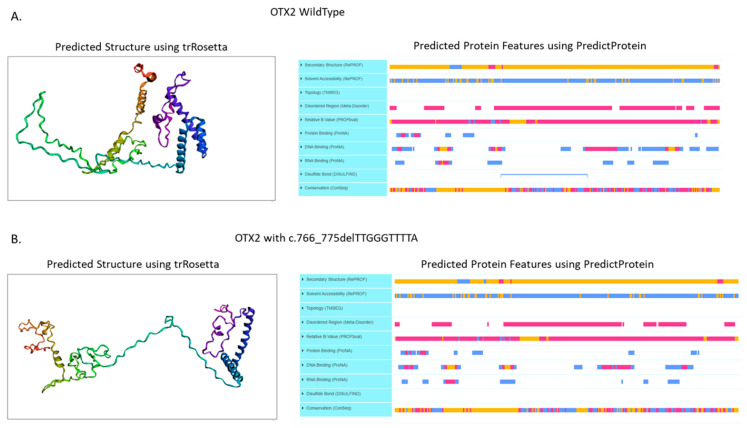
(**A**) Predicted structure and protein features of the wild type *OTX2* protein. (**B**) Predicted structure and protein features of the frameshift variant Leu256ThrfsTer43 on the *OTX2* protein. Predicted structure was obtained using the trRosetta tool (https://yanglab.nankai.edu.cn/trRosetta/ access on 15 November 2022), instead the predicted protein features were calculated using the PredictProtein tool (https://predictprotein.org/ access on 15 November 2022).

## Data Availability

Not applicable.

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
