# Peer review of "Agnathia-Otocephaly Complex Due to a De Novo Deletion in the OTX2 Gene"

_genes, 2022, doi:10.3390/genes13122269_

Round 1

Reviewer 1 Report

Reviewer 1

General comment:

The manuscript of Marco Fabiani and colleagues report a case of a fetus with severe Agnathia-otocephaly complex (AOC) diagnosed on routine ultrasound scan in the first trimester of pregnancy. The authors found a novel de novo variants in the OTX2 gene, never been reported before. The possibility of a correlation between the genotype of orthodenticular homeobox 2 (OTX2) and the phenotype of the studied pathology was shown.

AOC remains a rare malformation and this report contributes to understanding its etiology and underlines the importance of reporting unusual variant chromosomes for diagnostic genetic purposes.

The text is written in good and understandable English using correct scientific terminology. All the relevant details are included in the case study. The discussion covers all important aspects of the case with its interpretation in the context of the available literature.

The design of the study is logical and meets the purpose, all methods give clear and well-explained results.

Overall Recommendation: Accept after minor revision (corrections to minor methodological errors and text editing)

However, there are some caveats where corrections and / or additions would be desirable to improve the publication.

Additional changes and clarifications in the text of the publication:

Line 89: 2.2. Gentetic Analysis.  - typo in title

Line 125 ACMG Classification - This classification needs to be deciphered, and it is necessary to provide links to this database

it is necessary to provide links to all programs and databases used in the study:

Line 109 Integrative Genomics Viewer (IGV) software

Line 124 ClinVar Database - needs to provide link to the program

Line 128 mutation pathogenicity prediction software (Mutation Taster, Polyphen2, Predict SNP, FATHMM, PhD-SNP and CADD) - needs to provide links to these programs

Line 134 (Clinvar, Varsome, 1000 Genome Project, HGMD, ExAC, and gnomAD). - needs to provide links to these programs

Line 141 predicted by DISULFIND - needs to provide link to the program

ambiguity can be traced here - it describes that:’’ that predicted structure of OTX2 protein bearing 140 c.766_775delTTGGGTTTTA deletion cause a loss of Disulfide bond (predicted by DISULFIND) (Figure 5A).  In this case, it is need to specify the program in the annotation to the figure.

Line 272-274 Detected variants were annotated and filtered based on information of functional prediction (e.g., Polyphen2, SIFT, REVEL), disease as sociation (e.g., ClinVar, HGMD, OMIM and GWAS) and population allele frequency (e.g., dbSNPs , ALFRED). - needs to provide links to these programs

Line 259 DNeasy Blood & TissueKit and QIAamp  and DNA Blood Mini Kit according to the manufacturer’s instructions. - manufacturer must be specified.

Line 234, 235 (predicted by DISULFIND) (Figure 4B) causing a probably stretching of protein structure (Figure 4A).   – Figure numbers need to be corrected

Line 305 The study was approved by the local ethical committee of Artemisia SPA - The identification code and the date of approval must be specified.

Author Response

Reviewer 1

General comment:

The manuscript of Marco Fabiani and colleagues report a case of a fetus with severe Agnathia-otocephaly complex (AOC) diagnosed on routine ultrasound scan in the first trimester of pregnancy. The authors found a novel de novo variants in the OTX2 gene, never been reported before. The possibility of a correlation between the genotype of orthodenticular homeobox 2 (OTX2) and the phenotype of the studied pathology was shown.

AOC remains a rare malformation and this report contributes to understanding its etiology and underlines the importance of reporting unusual variant chromosomes for diagnostic genetic purposes.

The text is written in good and understandable English using correct scientific terminology. All the relevant details are included in the case study. The discussion covers all important aspects of the case with its interpretation in the context of the available literature.

The design of the study is logical and meets the purpose, all methods give clear and well-explained results.

Overall Recommendation: Accept after minor revision (corrections to minor methodological errors and text editing)

However, there are some caveats where corrections and / or additions would be desirable to improve the publication.

Additional changes and clarifications in the text of the publication:

Line 892.2. Gentetic Analysis.  - typo in title Done

Line 125 ACMG Classification - This classification needs to be deciphered, and it is necessary to provide links to this database

Thank you for the comment. A reference about ACMG classification was added to clarify the code. We include in the manuscript also indication where ACMG classification of variant can be found.

it is necessary to provide links to all programs and databases used in the study:

Line 109 Integrative Genomics Viewer (IGV) software

Line 124 ClinVar Database needs to provide link to the program

Line 128 mutation pathogenicity prediction software (Mutation Taster, Polyphen2, Predict SNP, FATHMM, PhD-SNP and CADD) needs to provide links to these programs

Line 134 (Clinvar, Varsome, 1000 Genome Project, HGMD, ExAC, and gnomAD). - needs to provide links to these programs

Line 141 predicted by DISULFIND needs to provide link to the program

ambiguity can be traced here - it describes that:’’ that predicted structure of OTX2 protein bearing 140 c.766_775delTTGGGTTTTA deletion cause a loss of Disulfide bond (predicted by DISULFIND) (Figure 5A).  In this case, it is need to specify the program in the annotation to the figure.

DISULFIND is a server and we have accessed to it by ProdictProtein tool. We explain better this aspect and links of both of them were added in the manuscript.

Line 272-274 Detected variants were annotated and filtered based on information of functional prediction (e.g., Polyphen2, SIFT, REVEL), disease as sociation (e.g., ClinVar, HGMD, OMIM and GWAS) and population allele frequency (e.g., dbSNPs , ALFRED). - needs to provide links to these programs

links to all programs and databases were now provided in the manuscript

Line 259 DNeasy Blood & TissueKit and QIAamp  and DNA Blood Mini Kit according to the manufacturer’s instructions. - manufacturer must be specified.

Manufacturer and link of protocol were provided

Line 234, 235 (predicted by DISULFIND) (Figure 4B) causing a probably stretching of protein structure (Figure 4A).   – Figure numbers need to be corrected

Thank you for the correction

Line 305 The study was approved by the local ethical committee of Artemisia SPA - The identification code and the date of approval must be specified.

The identification code and the date of approval were reported in the manuscript

Reviewer 2 Report

This is an interesting report presented by authors on a very rare disorder. However, I will suggest that please use OMIM number of disease and genes. 

Also please see the https://www.genenames.org/about/guidelines/. And italicized all Human genes 

Author Response

Reviewer 2

This is an interesting report presented by authors on a very rare disorder. However, I will suggest that please use OMIM number of disease and genes. 

Thank you for your suggestion OMIM numbers of diseases and genes were included.

Also please see the https://www.genenames.org/about/guidelines/. And italicized all Human genes 

Thank you for your suggestion all genes were italicized
